# Ventral Morphology of the Non-Trilobite Artiopod *Retifacies abnormalis* Hou, Chen & Lu, 1989, from the Early Cambrian Chengjiang Biota, China

**DOI:** 10.3390/biology11081235

**Published:** 2022-08-19

**Authors:** Maoyin Zhang, Yu Liu, Xianguang Hou, Javier Ortega-Hernández, Huijuan Mai, Michel Schmidt, Roland R. Melzer, Jin Guo

**Affiliations:** 1Key Laboratory for Palaeobiology, Institute of Palaeontology, Yunnan University, North Cuihu Road 2, Kunming 650091, China; 2MEC International Joint Laboratory for Palaeobiology and Palaeoenvironment, Yunnan University, North Cuihu Road 2, Kunming 650091, China; 3Chengjiang Fossil Museum of the Management Committee of the Chengjiang World Heritage Fossil Site, Yuxi 652599, China; 4Museum of Comparative Zoology and Department of Organismic and Evolutionary Biology, Harvard University, Cambridge, MA 02138, USA; 5Bavarian State Collection of Zoology, Bavarian Natural History Collections, Münchhausenstr. 21, 81247 Munchen, Germany; 6Department Biology II, Ludwig-Maximilians-Universität München, 82152 Planegg-Martinsried, Germany; 7GeoBio-Center, Ludwig-Maximilians-Universität Munich, Luisenstr. 37, 80333 München, Germany

**Keywords:** ventral morphology, Cambrian artiopoda, micro computed tomography, Trilobitomorpha

## Abstract

**Simple Summary:**

This study reveals the detailed ventral morphology of the enigmatic Cambrian non-biomineralized euarthropod *Retifacies abnormalis* from the early Cambrian of South China through the use of micro-computer tomography and three-dimensional modelling. The ventral morphology of *R. abnormalis* includes five pairs of cephalic appendages in three forms (one uniramous antenna pair, three uniramous appendage pairs, one biramous appendage pair), biramous trunk appendages with multilamellar exites, a variable number of pygidial appendage pairs (five or six) and a multi-articulated tailspine with two pairs of short lobe-like accessories. The new anatomical data inform the ecology and evolution of *R. abnormalis* in the broader evolutionary context of trilobite-like euarthropods known from sites of exceptional preservation. *R. abnormalis* possessed a higher degree of appendage differentiation along the body than initially thought, in parallel to similar discoveries of other trilobitomorphs such as *Naraoia spinosa*, *Pygmaclypeatus daziensis*, and *Sinoburius lunaris*. This discovery provides additional support to the hypothesis that early trilobitomorphs were ancestrally characterized by heteronomous ventral appendages with various degrees of functional specialization for feeding and respiration.

**Abstract:**

The artiopodans represent a diverse group of euarthropods with a typically flattened dorsal exoskeleton that covers numerous pairs of biramous ventral appendages, and which are ubiquitous faunal components of the 518-million-year-old Chengjiang Lagerstätte in South China. Despite their abundance, several Chengjiang artiopodans remain poorly known, such as the large euarthropoda *Retifacies abnormalis,* Hou, Chen & Lu, 1989, which is distinguished by the presence of mesh-like ornamentation on its dorsal exoskeleton. Although only a few ventral details were described in a single study in 25 years, it has been frequently featured in phylogenetic analyses that explore the relationships between Cambrian euarthropods. Here, we employ micro-CT and fluorescent microphotography to investigate the exceptionally preserved ventral morphology of *R. abnormalis* and explore its phylogenetic implications through maximum parsimony and Bayesian inference. Detailed morphology revealed here better supports *R. abnormalis* as a sister group to the diminutive artiopod *Pygmaclypeatus daziensis*, also known from Chengjiang, and strengthens the close relationship of these taxa that have been suggested by previous studies as early-branching representatives of Trilobitomorpha. Cephalic appendages suggest this animal might be a scavenger, possibly feeding on soft-bodied organisms. Different pairs of pygidial appendages suggest an anamorphic post-embryonic ontogeny, which adds to the understanding of the developmental mode of Cambrian artiopods, and further supports the statement that post-hatching segment addition occurred in the ancestor of Euarthropoda.

## 1. Introduction

Artiopoda [1] represents a diverse group of Palaeozoic euarthropods with a worldwide distribution that includes biomineralized forms such as trilobites, as well as non-biomineralized representatives that are exclusively known from sites of exceptional preservation [2,3,4,5,6]. Most artiopods are characterized by a dorsal exoskeleton with well-developed pleurae that convey a flattened appearance, and a consistent appendage organization including a uniramous set of antennae, followed by a variable number of typically biramous appendages composed of a gnathobasic protopodite, an endopodite that functions as a walking leg, and a lamellae-bearing exopodite. Although earlier work regarded artiopod appendages as nearly homonomous in their construction (e.g., [1,4]), discoveries have demonstrated an important degree of limb heteronomy and functional differentiation in various representatives. Recent trilobite studies have shown differences between the organization of the anterior and posterior appendages [7], tested the biomechanical performance of different gnathobasic protopodite morphologies [8], and even demonstrated the existence of sexual dimorphism expressed as limbs modified into male claspers [9]. However, some of the most striking examples of limb differentiation within individual specimens are found in non-trilobite artiopods from the early Cambrian Chengjiang biota in South China, such as the xandarellid *Sinoburius lunaris* [10], and the early-branching artiopod *Pygmaclypeatus daziensis* [11]. These organisms show evidence of drastically modified exopodite and endopodite morphologies along the body—often expressed between the head and trunk regions—and indicate a degree of functional differentiation for food processing that extends beyond the traditional view of these organisms as mud-feeders or simple scavengers (e.g., [1]). Much of this progress has been possible thanks to the availability of new material, coupled with the application of X-ray-based micro computed tomography (micro-CT) that facilitates the study of pyritized Chengjiang fossils that are problematically small, rare, or incompletely preserved for conventional light photography [10,11,12]. 

In this contribution, we apply micro-CT to redescribe the ventral morphology of the non-trilobite artiopod *Retifacies abnormalis* Hou, Chen & Lu, 1989, from Chengjiang, and explore its significance against the evolution of Cambrian euarthropods. *R. abnormalis* is a large artiopod whose most striking characteristic is the presence of a heavily ornamented dorsal exoskeleton consisting of a dense reticulate pattern, as well as the rare combination of a prominent pygidial shield and a multi-articulated tailspine [1,13]. Although initially described by Hou [14] based on the dorsal exoskeleton only, the ventral organization of *R. abnormalis* has only been addressed in detail by Hou and Bergström (1997) [1] (pp. 52–57) based on new material at the time, including several specimens with well-preserved appendages including elongated uniramous antennae and biramous appendages, as well as the distinctive multi-articulated tailspine. The former study included specimens of *R. abnormalis* with a maximum length (sagittal) for the articulated dorsal exoskeleton only of 35 cm and an estimate of up to 55 cm, including the fully extended antennae and tailspine (1997) [1] (pp. 52–57). Despite the availability of new figured specimens (e.g., [13] (pp. 194–195, Figure 20.27)), the morphology of *R. abnormalis* has not been formally redescribed for almost 25 years, despite the fact that this taxon is frequently included in phylogenetic analyses aimed at reconstructing the phylogeny of Cambrian euarthropods. *R. abnormalis* has been regarded as a relative of the concilitergans *Helmetia* and *Kuamaia* [15], whereas Hou and Bergström [1] (p. 43) suggested a closer relationship with *Naraoia* as part of their subclass Nectopleura based on the presence of a large pygidial shield, limbs with wrinkled proximal basis, ventral stalked eyes, and feeding through mud ingestion. Phylogenetic analyses have instead recovered *R. abnormalis* as a member of Trilobitomorpha with uncertain affinities (e.g., [2]), as the sister group to all other trilobitomorphs (e.g., [3,5,10,16]) or as artiopods of uncertain affinity (e.g., [6]). Although recent studies have consistently suggested a sister group relationship between *R. abnormalis* and the rare Chengjiang euarthropod *Pygmaclypeatus daziensis* based on the shared presence of a pygidium and a multi-articulated tailspine in both taxa, the precise position of this small unnamed clade relative to Trilobitomorpha remains sensitive to different search parameters (see [3,11,17]). Here, we explore the implications of the new information on the ventral morphology of *R. abnormalis* to better understand the relationships and macroevolution of Cambrian artiopods.

## 2. Materials and Methods

Ninety-six specimens were stored at the Yunnan Key Laboratory of Paleobiology. Specimens were collected from the Yu’anshan Member, Chiungchussu Formation, *Eoredlichia*-*Wutingaspis* trilobite biozone, Cambrian Series 2, Stage 3, Ercaicun, Yunnan Province, China. Nine specimens preserved sufficient details to merit being photographed with a camera or scanned with micro-CT (see below). YKLP 13926 and YKLP 13927 of these were the basis of pp. 194–195 in study [13], which provided some details of antennae and tailspine. 

Macro-photography of the fossil material was undertaken using a Canon EOS 5DS R camera (DS126611, Canon Inc., Tokyo, Japan) with MACRO PHOTO LENS MP-E 65 mm, with directional illumination provided by a LEICA LED5000 MCITM (Leica Microsystems, Wetzlar, Germany). Images were often recorded as a stack at different focal planes, especially for enlargements. Fluorescence microscopic images were captured with a Leica DFC7000T CCD linked to a Leica M205 FA fluorescent microscope (Leica Microsystems, Wetzlar, Germany). Computed tomography was performed on a Zeiss Xradia 520 Versa (Carl Zeiss X-ray Microscopy, Inc., Pleasanton, CA, USA) for 13 specimens including YKLP 13926 and YKLP 13927; however, only four specimens preserved fine details. The four specimens were YKLP 11430 (tomography resolution ranged from 7.39 to 35.92 μm), YKLP 11426 (tomography resolution ranged from 17.00 to 27.45 μm), YKLP 11432 (tomography resolution ranged from 13.31 to 19.83 μm), and YKLP 11436 (tomography resolution 18.03 μm). The resolutions acquired depended on the size of the scanned region and the slab. For further details on parameters see Table 1. A set of radiographs generated from each scan were saved as TIFF stacks and processed with Drishti 2.4 [18]. Images were colour- and contrast-optimized and arranged into figures in Adobe Photoshop CC 2018.

## 3. Results

### 3.1. Systematic Palaeontology

Euarthropoda Lankester, 1904 [19]; 

Artiopoda Hou & Bergström, 1997 [1];

Genus Retifacies Hou, Chen & Lu, 1989 [14].

#### 3.1.1. Emended Diagnosis

Large artiopod with dorsoventrally flattened dorsal exoskeleton, longer than wide, with widespread reticulated ornamentation. Dorsal exoskeleton with weakly defined axial region, divided into the cephalic shield, thorax with 10 freely articulating tergites, and pygidial shield. The cephalic shield is wide and short, covering a conterminant hypostome, one pair of uniramous antennae, and four pairs of cephalic ventral appendages consisting of three uniramous sets anteriorly, and one biramous set posteriorly. Each thoracic tergite covers one pair of ventral biramous appendages, whereas pygidium covers up to six pairs of biramous appendages and an elongated multi-articulated tailspine. Cephalic uniramous appendages consist of an endopod with seven podomeres, including an elongated and gently curved terminal claw. Protopodite morphology in cephalic appendages uncertain. All biramous appendages (from last cephalic appendage to last pygidial appendage) consist of a subrectangular smooth protopodite, endopod with six podomeres with robust paired endites, an exopod with a distal lobe bearing paddle-shaped lamellae, and a proximal exite with lamellae. Pygidial shield macropygous, with a single pair of short posterior-facing spines. Modified from [1,13,14].

#### 3.1.2. Remarks

*Retifacies abnormalis* is exclusively known from the Chengjiang biota in South China [13], and represents the only species described for this genus to date [14]. Luo et al. (1997) suggested the species *Retifacies longispinus* Luo & Hu 1997 based on the presence of a curved caudal spine that projects out from underneath the pygidium [20], but they later considered *R. longispinus* as a synonym of *R. abnormalis* [21]. Hou and Bergström proposed the Order Retifaciida and the Family Retifaciidae, and subsequently proposed that the enigmatic artiopod *Squamacula clypeata* Hou & Bergström, 1997, might represent a member of this family based mainly on the presence of a thorax with 10 freely articulating tergites and the interpretation that the cephalic shield covers four pairs of appendages, including the antennae [1] (pp. 52–60). However, both *R. abnormalis* and *S. clypeata* have been included in numerous cladistic analyses, none of which resolve a close relationship between these taxa that would support that they belong to a monophyletic group (e.g., [3,5,6,11,17,22]). Given the lack of phylogenetic support for Retifaciidae, we avoid the use of these names in a formal systematic treatment given that the precise position of *R. abnormalis* and its close relatives remain somewhat fluid within the context of early branching Artiopoda.

The revised morphology of *R. abnormalis* made possible by the use of micro-CT techniques indicates several differences relative to the original description. We find no evidence for stalked eyes (*contra* the suggested presence of eyes based on a single specimen figured by Hou and Bergström [1] (p. 56, Figure 51)). We identify the presence of four pairs of cephalic appendages consisting of three uniramous and one biramous (contra interpretation for three pairs of biramous cephalic appendages [1,13]). We find evidence for up to six pairs of biramous appendages underneath the pygidial shield. The structure of the ventral appendages includes a protopodite, exopodites with densely packed lamellae, and endopodites with six podomeres, with paired endites on the thorax and pygidium, including a well-developed and gently curved terminal claw (*contra* seven podomeres and terminal claw, all of which were interpreted as featuring up to five ventral spinose endites each [1,13]). We also find new information on the morphology of the protopodite associated dorsally with lamellate exites [1] (pp. 52–57).

*Retifacies abnormalis* Hou, Chen & Lu, 1989 [14].

1989. Hou, Chen & Lu, pl, Figures 1–6; pl. II, Figures 1–4; text-Figures 1 and 2 [14]. 

1997. Hou & Bergström, pp. 52–57, Figures 48–52 [14].

1997. Luo et al., Plate II. 5, 6 [20].

1999. Luo et al., p. 46, Plate II, Figures 2–5 [21].

2017. Hou et al., pp.194–195, Figures 20.26 and 20.27 [13].

Diagnosis: As for genus.

### 3.2. Description

#### 3.2.1. Preservation and Overall Morphology 

Most of the studied specimens were preserved with the exoskeleton in dorsal view (Figure 1, Figure 2, Figure 3, Figure 4C,D, Figure 5A,C and Figure A1B,D,E, Appendix A), most likely due to the flattened overall organization of the body. Only one specimen was buried in a lateral orientation (Figure 4A,B). Specimens (>100) were collected at the Yunnan Key Laboratory for Palaeobiology, Institute of Palaeontology, Yunnan University in China.

Range in length (sagittal) from 29.86 to 94.88 mm, excluding the antennae and tailspine, and maximum width from 16.65 to 72.07 mm was measured at the fifth thoracic tergite (Table A1). The specimen with five pairs of biramous appendages beneath the pygidium (YKLP 11430) was 41.32 mm long and about 30 mm wide (Figure 1E and Figure 2A, Appendix A), while the specimens with six pairs of appendages beneath the pygidium were larger (specimen YKLP 11432 was 51.06 mm long and 33.03 mm wide, length of specimen YKLP 11426 was unknown, and its width was about 23 mm). 

The dorsal exoskeleton is sub-elliptical in overall outline, and can be divided into a cephalic shield, a thorax with 10 freely articulating tergites, and a pygidium (Figure 6A,B). The dorsal side of the exoskeleton has a poorly defined raised axial region without furrows and is heavily ornamented with a reticulated pattern consisting mostly of five- and six-sided polygons. The exoskeleton of a whole animal divides into a head shield, ten thoracic tergites, and a pygidium, making up about 8–16%, 53–64%, and 22–34% of the total length, respectively (see Table A1).

**Figure 1 biology-11-01235-f001:**
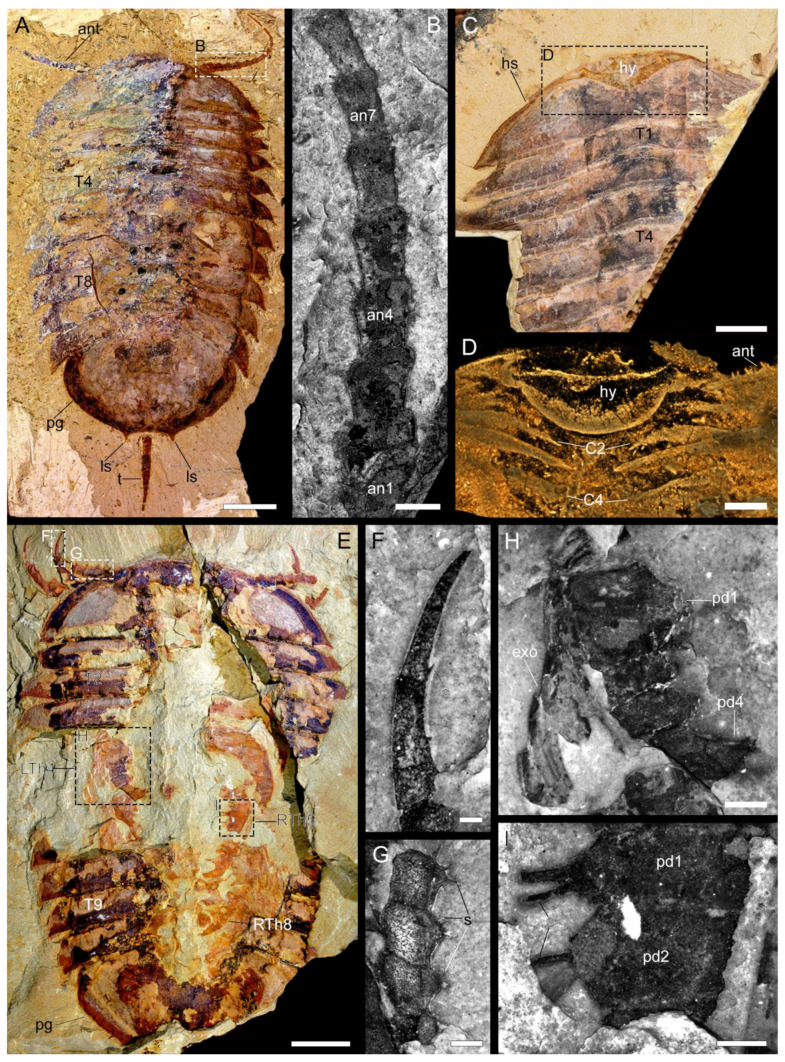
Macro-photographic (**A**,**C**,**E**), fluorescent (**B**,**F**–**I**), and tomographic (**D**) images of *Retifacies abnormalis*. (**A**) Dorsal view of YKLP 13927. (**B**) Close-up of the proximal eight annuli of right antenna in (**A**). (**C**) Dorsal view of YKLP 11436. (**D**) Head region shown in (**C**) demonstrates the morphology and position of the hypostome relative to the cephalic appendages. (**E**) Dorsal view of YKLP 11430. (**F**) Distal portion of the first left post-antennal appendage. (**G**) Close-up of left antenna in (**E**). (**H**) Close-up of the fourth left thoracic appendage (LTh4) with endopodite and exopodite. (**I**) Close-up of two podomeres of the fifth right thoracic appendage (RTh5). Note the prominent enditic spines. Abbreviations: ant—antenna (e); ann—nth annulus of the antenna; Cn—nth post-antennal cephalic appendage; exo—exopodite; hs—head shield; ls—lateral spines on the posterior margin of the pygidium; pdn—nth podomere of the aapendages; pg—pygidium; s—spine (s)/seta (e); Tn—nth thoracic tergite; Thn—nth thoracic appendages; tc—terminal claw; R—right; L—left. Scale bars: 1 cm in (**A**), 5 mm in (**C**,**E**), 2 mm in (**D**), 1 mm in (**B**,**H**), 0.5 mm in (**G**,**I**), and 0.2 mm in (**F**).

#### 3.2.2. Cephalon

*Cephalic shield*. Corresponds to approximately 8–16% of the total length (sag.) of the dorsal exoskeleton (Table A1). The cephalic shield is broad and short, approximately 4.5 times wider (trans.) than long (sag.) conferring a semi-elliptical overall appearance (Figure 1A,C,F, Figure 2A, Figure 4A and Figure A1B, Appendix A). The antero-lateral margins of the cephalic shield are slightly straight, rather than curved, and extend into well-defined but small genal spines that point postero-laterally. The posterior margin of the cephalic shield is slightly anteriorly recurved. There is no evidence for any dorsal morphological features (e.g., glabella, eyes), with the exception of the reticulate ornamentation consisting of an irregular mesh of pentagonal or hexagonal polygons. The reticulation on the cephalic shield is finer compared to that of other parts of the body.

The ventral side of the head shows the preservation of a conterminant hypostome attached to the anterior margin of the cephalic shield in two specimens (YKLP 11436, Figure 1C,D, Appendix A; YKLP 11433, Figure A1B,C). The hypostome has a semi-circular shape, reaches posteriorly to approximately 50% of the total length (sag.), and covers 25% of the maximum width (trans.) of the cephalic shield. The hypostome overlaps with the base of the antennae and the first post-antennal cephalic appendage (Figure 1C,D, Appendix A). We found no evidence for ventral stalked eyes in any of the studied specimens (Figure 1A,C,D,F, Figure 2A, Figure 4A and Figure A1B, Appendix A).

*Cephalic appendages*. The first appendage pair corresponds to elongated and multi-articulated antennae that extend beyond the anterior margin of the cephalic shield (Figure 1A,E and Figure 2A, Appendix A). The antennae are well preserved in YKLP 13927 and show the presence of approximately 17 articles that become more elongated (sag.) and narrow (trans.) towards their distal ends. The articles have up to three endite-like spines located close to the anterior segmental margin on the ventral side of the antennae, which are particularly well-expressed on the proximal articles (Figure 1A,B,E,G, Figure 2A,B,C and Figure A1A, Appendix A). The antennal articles telescope into each other, with the widest (trans.) portion expressed on the most distal half that bears the endite-like spines (Figure 1A,B,E,G and Figure A1A). 

The cephalic shield covers four pairs of tightly packed post-antennal appendages (Figure 2A,B,C, Appendix A). The first three pairs of appendages lack an exopodite or exites, and only the fourth pair shows clear evidence for the presence of a well-defined exopodite (Figure 2A,B,C, Appendix A). The first three pairs of uniramous post-antennal appendages share the same morphology consisting of well-developed endopodites with approximately six podomeres and a strongly curved terminal claw; despite the good quality of preservation and use of micro-CT, details of the exact podomere boundaries are uncertain due to the extent of pyritization. The podomeres on the proximal half of the endopodite have a subquadrate shape and lack endites, whereas the distal podomeres have a more elongated sub-rectangular outline and bear well developed distal-facing endites on their anterior margins. The terminal claw is longer than the two preceding podomeres, has a strong acute and gently curved tip, and bears a single short spine along its median portion (Figure 1E,F). The fourth pair of post-antennal appendages is biramous as indicated by the presence of an exopodite. The endopodite has a similar morphology to those of the three anterior appendages, including the presence of an elongated and acute terminal claw (Figure 1A,B,C and Figure 6A–C,E,F). 

**Figure 2 biology-11-01235-f002:**
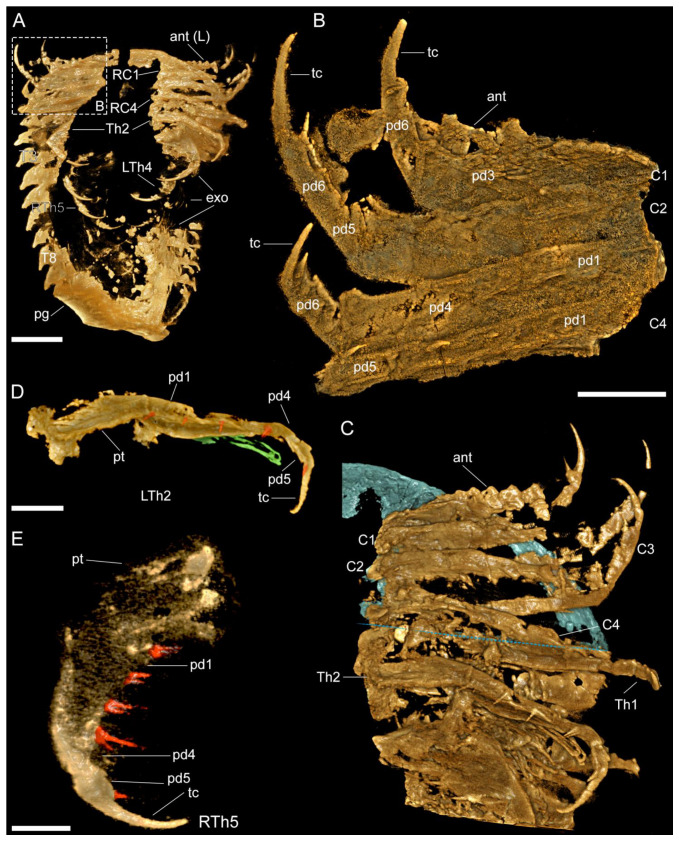
Images of ventral view rendered from 3D tomographic models of *Retifacies abnormalis* (YKLP 11430, Figure 1E). (**A**) Overview. (**B**) Close-up of right cephalic appendages (head shield removed). (**C**) Close-up of left cephalic appendages, with head shield false-coloured in blue, and the posterior edge of head shield marked with a blue–dashed line. (**D**) Close-up of second left thoracic appendage (LTh2 in (**A**,**C**)) with exopodite false-coloured in green and enditic setae false-coloured in red. (**E**) Close up of fifth right thoracic appendage (RTh5) with enditic setae false-coloured in red. Other abbreviations; pt—protopodite. Scale bars: 1 cm in (**A**), 2 mm in (**B**–**D**), and 1 mm in (**E**).

**Figure 3 biology-11-01235-f003:**
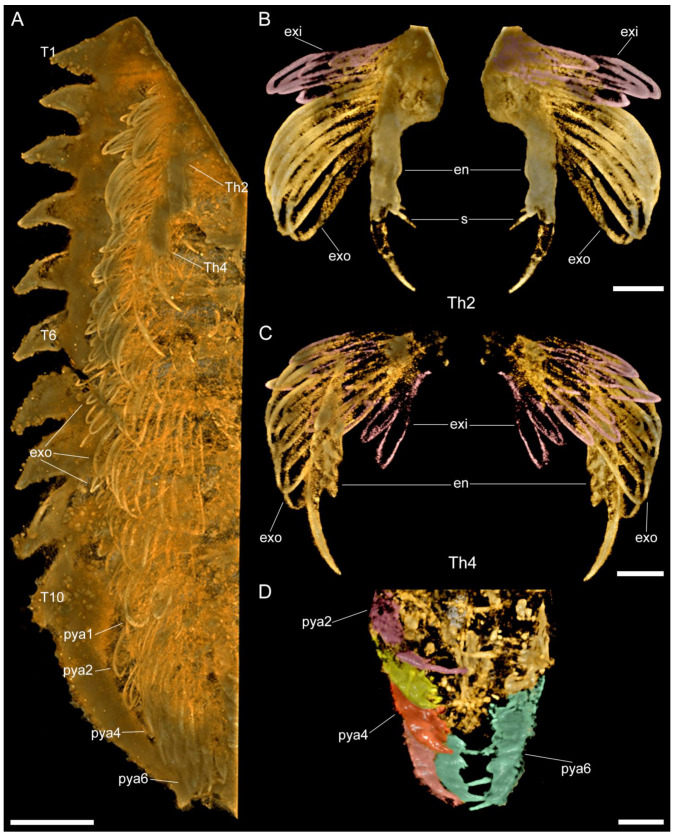
Images rendered from 3D tomographic models of *Retifacies abnormalis* (YKLP 11426, Figure A1D) showing details of thoracic and pygidial appendages. (**A**) Ventral view of appendages from the right side of the animal. (**B**,**C**) Ventral and dorsal views of second (Th2) and fourth (Th4) right thoracic appendages with exites (false-coloured in purple) proximally attached to their respective protopodites. (**D**) Ventral view of endopodites of all preserved pygidial appendages, with pygidial shield and exopodites digitally removed for clarity. Each endopodite false-coloured differently. Other abbreviations: en—endopodite; exi—exite; pyan—nth pygidial appendage. Scale bars: 5 mm in (**A**) and 2 mm in (**B**–**D**).

#### 3.2.3. Thoracic tergites and Appendages

The thorax is composed of 10 freely articulating and overlapping tergites and corresponds to 53–64% of the total body length (sag.) (Table A1). All thoracic tergites are nearly identical in their overall morphology and length (sag.), and only vary slightly in their maximum width, with the fifth tergite being the widest, and the degree of curvature. The thoracic tergites have well-developed tergopleurae that match the width (trans.) of the cephalic shield. The anterolateral edge of the tergites is gently curved, and develops into a well-defined but short pleural spine abaxially that faces postero-laterally. The posterior margin of the tergites is almost straight, except for a gentle anteromedian curvature that defines a weak raised axial region throughout the trunk. The thoracic tergites become progressively more recurved towards the posterior end of the body, with the first to third tergites having a straight profile, whereas the ninth and tenth tergites have a broad crescent shape (Figure 1A,C,E and Figure A1B,D,E). Each tergite has two rows of mesh-like ornaments, and the meshes of the anterior row are large and somewhat rectangular, while the posterior ones are small and nearly square-shaped (Figure 1A,C,E and Figure A1D) (see also [14] their plates I and II; [1] their Figures 48B and 50A; [13] their Figures 20.26 and 20.27c).

Each thoracic tergite covers a pair of biramous appendages composed of an enlarged protopodite (Figure 1F,H,I), an endopodite with six podomeres, and a lamellae-bearing exopodite (Figure 1F,H). The enlarged protopodite has a subrectangular shape, with a transversely elongated profile relative to its corresponding endopodite, and notably has a smooth ventral edge (Figure 1E,H,I, Figure 2D,E and Figure 3B,C, Appendix A). Dorsally, the protopodite carries a well-developed dorsal exite that consists of several lamellae that originate from the proximal portion of the biramous appendage (Figure A2B) (see also [12]). The endopodite has five podomeres and a well-developed terminal claw. Each podomere has a subrectangular outline that is wider (dorsoventrally) than it is long (sag.) and carries a pair of spinose endites on the ventral side (Figure 1H,I, Figure 2D,E and Figure 3B). The prominent terminal claw is nearly as long as the two preceding podomeres, and has a curved outline and an acute termination (Figure 1E,F, Figure 2D,E and Figure 3B,C, Appendix A). The exopodite consists of a single semicircular lobe that bears over a dozen paddle-shaped lamellae, from which only the distal-most lamella carries a series of fine marginal setae (Figure 2A, Appendix A). All thoracic biramous appendages share the same overall organization (Figure 6A–C,F).

#### 3.2.4. Pygidium and Pygidial Appendages

The pygidial shield is large, encompassing 22–34% of the total body length (sag.) (Table A1). The pygidium has an overall subtrapezoidal outline, with gently curved lateral margins that develop into a pair of short posterior-facing spines, and a slightly anteriorly curved posterior margin. A multi-articulated and elongated tailspine extend below the posterior curved margin of the pygidium, in between the short spines. The tailspine is longer than half the thorax (sag.) and consists of at least a dozen quadrate-shaped articles of a similar length (sag.) that taper in width (trans.) towards the distal termination (Figure 4A,B, [1] (pp. 55–56, Figures 50 and 51)). Similar to the thorax, the pygidium is covered in a dense network of five- and six-sided polygons, with those closer to the margin being comparatively smaller to the larger ones found in the centre of this sclerite.

The pygidial shield overlies between five (observed in YKLP 11430) and six (observed in YKLP 11426 and YKLP 11432) pairs of biramous appendages with a similar overall morphology to those found in the thorax (Figure 3A, Figure 5A,C and Figure A1A, Appendix A). The difference in size between the specimens (YKLP 11430 body size of 41.32 mm long and 30 mm wide or so; YKLP 11426 body size of uncertain length and about 35 mm wide; YKLP 11432 body size of 51.06 mm long and 33.03 mm wide) suggests that the number of appendages under the pygidium might be a consequence of the formation of new segmental appendages during ontogenetic growth. The pygidial appendages are better preserved than those in the thoracic region in the studied specimens, and thus show the overall organization of the biramous limbs in detail, including the protopodite, exite, endopodite and exopodite lamellae (Figure 2, Figure 3, Figure 5, Figure 6 and Figure A2, Appendix A). Additionally, marginal spines preserved on YKLP 11430 are arranged on the distal lamella of each exopodite (Figure 1E, Figure 2A, Figure 5A and Figure A2A,C, Appendix A). On several specimens, the tailspine emerges ventrally, with its anterior end located posterior to the last pygidial appendage (Figure 1A, Figure 4A–D, Figure 5C and Figure A1D,E, Appendix A). Additionally, YKLP 11432 preserves two pairs of specialized accessories near the anterior end of the tailspine (Figure 4C,D, Figure 5C and Figure 6A,B,G, Appendix A) that have not been reported in any previous studies.

**Figure 4 biology-11-01235-f004:**
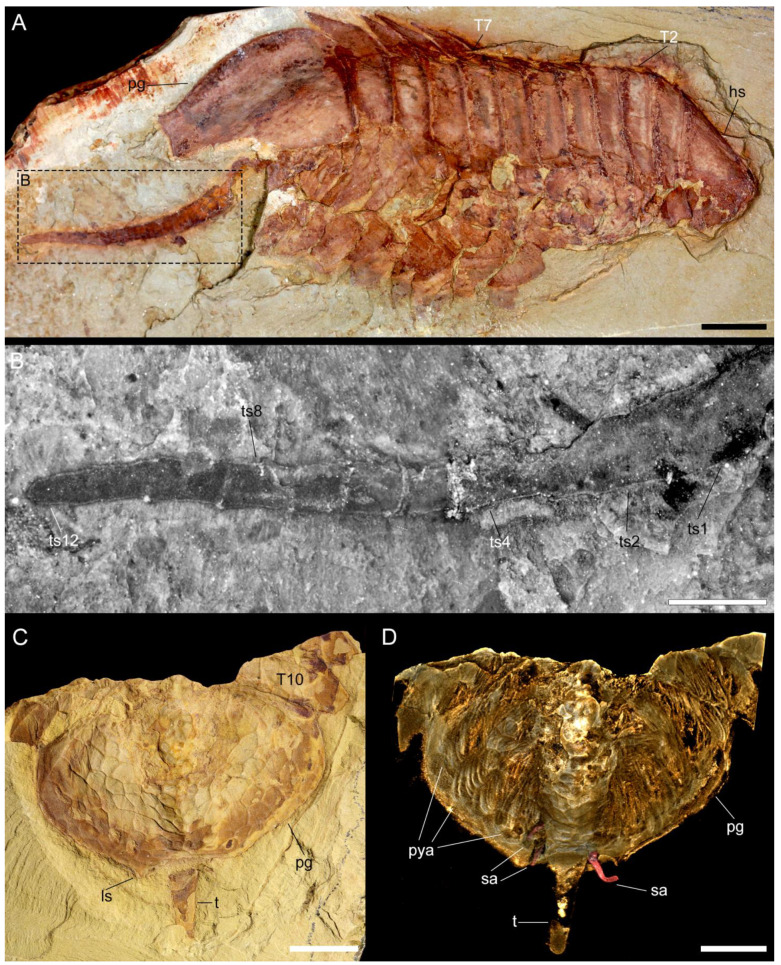
Macro-photographic (**A**,**C**), fluorescent (**B**), and tomographic (**D**) images of *Retifacies abnormalis* showing details of the tailspine and pygidium. (**A**) Dorsal view of CJHMD00025. (**B**) Close-up of tailspine in (**A**). (**C**) Dorsal view of YKLP 11432 showing mesh-like ornaments on the dorsal of pygidium. (**D**) Dorsal view of YKLP 11432 showing pygidial appendages and specialized accessories on the anterior end of tailspine. Other abbreviations: tsn—nth tailspine segment; sa—specialized accessories. Scale bars: 5 mm in (**A**,**C**,**D**) and 2 mm in (**B**).

**Figure 5 biology-11-01235-f005:**
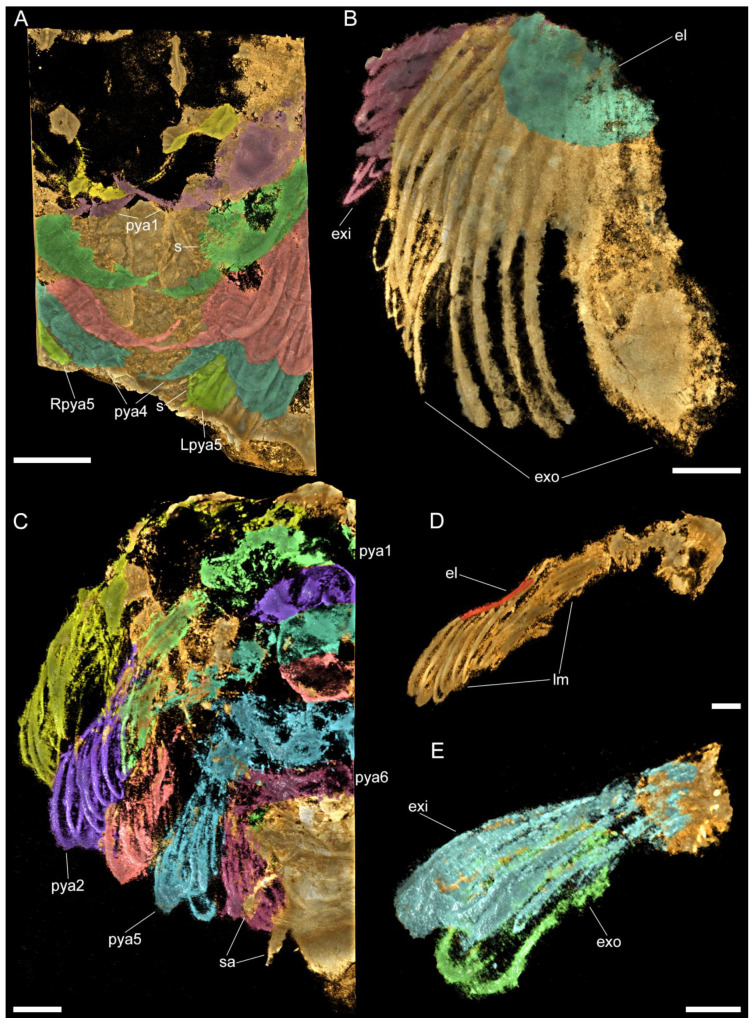
Images rendered from 3D tomographic models of *Retifacies abnormalis* ((**A**,**B**) rendered from YKLP 11430, see Figure 1E; (**C**–**E**) rendered from YKLP 11432, see Figure 4C). (**A**) Ventral view of pygidial appendages (each shown in a different false colour). (**B**) Ventral view of the exopodite and exite on the first pygidial appendage. Note the different directions and positions of the exite (purple) and the exopodite lamellae. Exopodite lobe (el) false-coloured in cyan. (**C**) Dorsal view includes all pygidial appendages from the left side of the animal. Lamellae of each appendage false-coloured differently. Pygidial shield completely removed for clarity. (**D**) Dorsal view of the second left pygidial appendage (pya2 in (**C**)) with exopodite lobe (el) false-coloured in red. (**E**) Dorsal view of the fifth left pygidial appendage with exites (exi) with lamellae false coloured in cyan and exopodites (exo) with lamellae false-coloured in green. Other abbreviations: lm—lamellae; sa—specialized accessories. Scale bars: 2 mm in (**A**,**C**) and 1 mm in (**B**,**D**,**E**).

**Figure 6 biology-11-01235-f006:**
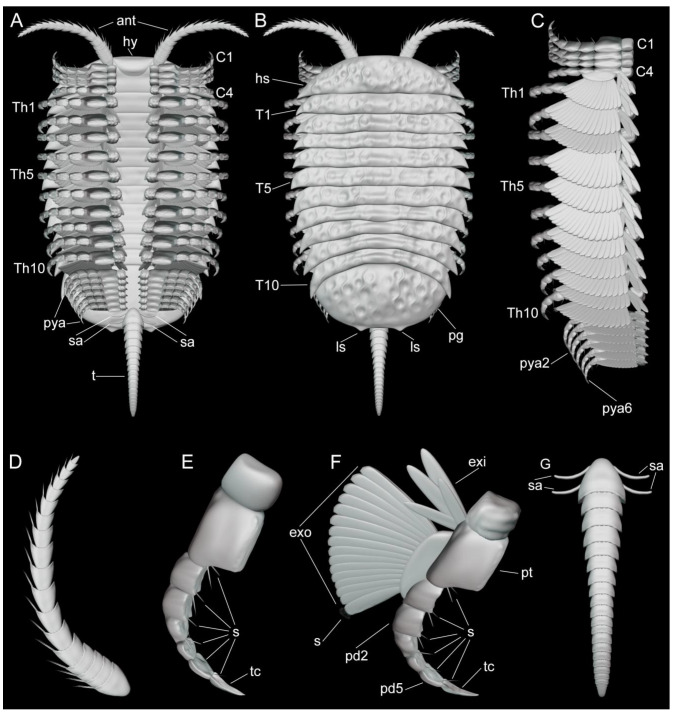
Three-dimensional morphological reconstruction of *Retifacies abnormalis*. (**A**) Ventral organization. (**B**) Dorsal view. (**C**) Dorsal view of post-antennal limbs (left) with dorsal shield removed to show their organization. (**D**) Antennae. (**E**) Morphology of 1st to 3rd post-antennal cephalic appendage pairs. (**F**) Morphology of the 4th cephalic appendage pair, ten thoracic appendage pairs and five or six pygidial appendage pairs. (**G**) Multi-segmented tailspine with two pairs of specialized accessories (sa).

### 3.3. Phylogenetic Analyses

We explored the phylogenetic implications of the new data on *R*. *abnormalis* using an updated version of the dataset by Schmidt et al. [11]. The character matrix consists of 65 taxa and 93 characters; we incorporated a new character, the presence of exites, based on the discovery of these structures in *R. abnormalis* and other euarthropods from Chengjiang [12]. Parsimony analyses were performed with TNT V1.5 [23] under New Technology Search, using Driven Search and Sectorial Search, Ratchet, Drift, and Tree fusing options activated with standard settings. The analysis was set to find the minimum tree length 100 times and to collapse trees after each search. All characters were treated as unordered. Analyses were performed under equal and implied weights. We also performed a Bayesian analysis in MrBayes 3.2 using the Monte Carlo Markov chain model for discrete morphological characters [24,25] for 1 million generations (four chains), with every 1000th sample stored and 25% burn-in. Convergence was diagnosed with the software Tracer [26], with effective sample size values over 200.

The phylogenetic analyses recovered a sister group relationship between *R*. *abnormalis* and the recently redescribed minute artiopod *Pygmaclypeatus daziensis*; these results were observed under equal weight parsimony (Figure 7A), implied weights (Figure 7B), and even Bayesian inference (Figure 7C). The relationship between these taxa has been recovered in previous studies (e.g., [3,11,17]), and thus further strengthens their position relative to each other. At a broader scope, *R. abnormalis* is recovered as an early branching member of Trilobitomorpha (Figure 7B), or alternatively in an unresolved polytomy outside this clade (Figure 7A,C). Although these results are sensitive to the type of analyses employed, they are largely consistent with previous studies have that addressed artiopod phylogenetic relationships and reflect the difficulty of reconstructing the basal interrelationships within this clade (see also [2,3,5,10,17]).

**Figure 7 biology-11-01235-f007:**
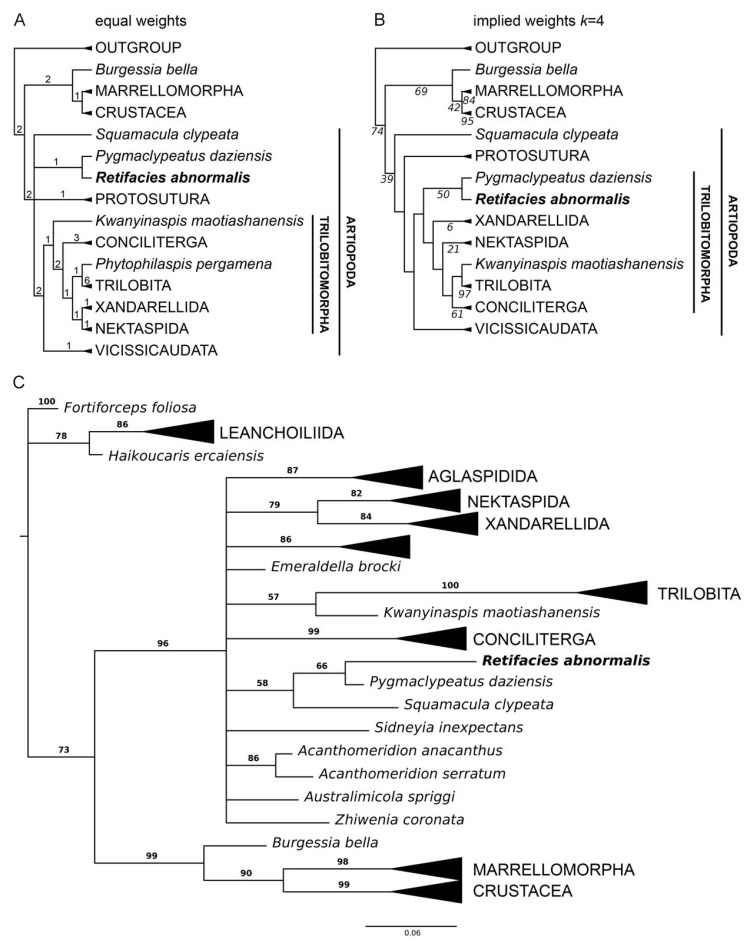
Phylogenetic position of *Retifacies abnormalis* within Artiopoda (Character matrix see the zip file Appendix A). (**A**) Strict consensus of the four most parsimonious trees under equal weights (271 steps; CI: 0.413; RI: 0.734); numerals indicate Bremer support values. (**B**) Single most parsimonious tree under implied weights (k = 4; CI: 0.404; RI: 0.724); italicized numerals indicate nodal support by frequency differences (GC) with symmetric resampling performed with 100 replicates. (**C**) Consensus tree resulting from Bayesian analysis in MrBayes. Mk model, four chains, 1,000,000 generations, 1/1000 sampling with 25% burn-in; bold numerals denote posterior probability values.

## 4. Discussion

The new morphological data on the ventral organization of *Retifacies abnormalis* leads us to produce a substantial update of this non-biomineralized artiopod. Eyes of *R. abnormalis* were not reported in the original description (see [14]), but were briefly described by Hou and Bergström based on one single specimen (CN 115388) using mechanical preparation [1] (their Figures 48A and 51), and have been considered as a valid character for this species in later publications (e.g., [2,10,11,13,22]). Unfortunately, CN 115388 is not available for the present study. None of the new specimens and micro-CT scans here reveal any structure similar to the putative eyes reported for CN 115388, which makes us question the interpretation for the existence of these ocular structures. We, therefore, consider *R. abnormalis* an eye-less artiopod, just as its close relative *Pygmaclypeatus daziensis* [11].

Other similarities shared by the head of *R. abnormalis* and *P. daziensis* include a conterminant hypostome, a pair of multi-segmented setose antennae extending beyond the anterior margin of the cephalic shield, and four pairs of post-antennal cephalic appendages. The only difference in the head organization of these two species seems to be the occurrence of highly reduced stenopodous exopodites associated with the first to third post-antennal biramous appendages in *P. daziensis*, whereas the corresponding appendages in *R. abnormalis* are uniramous. 

Despite the uniramous versus biramous organizations of the post-antennal appendages in the head of *R. abnormalis* and *P. daziensis*, the number (i.e., four pairs) of such cephalic appendages seems to remain stable not only for artiopods such as *Naraoia spinosa* (see [13]) and *Sinoburius lunaris* (see [10]), but also for megacheirans such as *Leanchoilia illecebrosa* (see [16] (their Figure S4)). Our observations argue against the notion that the euarthropod head with an eye segment, an antennal segment, and three segments with biramous limbs is one of the autapomorphies for the Euarthropoda (see [27] (their Figure 5)). The first post-antennal segment and its appendage are small and covered by the head shield and, therefore, could have been overlooked or inaccessible with conventional methods (see [27] for a similar situation reported in *L. illecebrosa*).

Compared to the uniramous head appendages, each with seven podomeres, the endopod of the last cephalic appendage and the thoracic and pygidial ones are composed of only six podomeres. Several thoracic and pygidial appendages are shown to bear a proximal exite composed of lamellae (Figure 5 and Figure A2A,B, Appendix A)—a feature recently reported for three other euarthropods from Chengjiang, namely, *Leanchoilia illecebrosa*, *L. obesa*, and *Naraoia spinosa* [12]. However, the presence of such an exite cannot be confirmed for the last cephalic appendage of *R. abnormalis*. This leads to a combination of four types of appendages on the ventral side of *R. abnormalis*: the uniramous antenna (one pair), the uniramous post-antennal cephalic appendages (three pairs), the biramous post-antennal cephalic appendage (one pair), and the biramous thoracic (10 pairs) and pygidial (five or six pairs) appendages (Figure 2, Figure 3, Figure 5, Figure 6 and Figure A2A,B, Appendix A) and marginal spines on the distal lamella of each exopodite of the fourth cephalic limb pair, thoracic limbs and pygidial limbs (Figure 2A, Figure 5A and Figure 6F, Appendix A). Again, this situation resembles that in *P. daziensis* [11], and is not yet known for any other artiopods. Our data confirm the close relationship between *R. abnormalis* and *P. daziensis* from the ventral morphology perspective.

*Tailspine and specialized accessories*. The last (i.e., fifth or sixth) pair of pygidial appendages are followed by a multi-articulated tailspine (Figure 1A, Figure 4, Figure 5C, Figure 6A,B,G and Figure A1D,E, Appendix A). The tailspine is attached to the ventral side of the pygidium and, therefore, is different from the dorsally located tailspines of, e.g., *Eoredlichia intermedia* [28], but similar to that of *P. daziensis* [11]. Given the segmentation and the enlarged proximal part of the tailspine, we speculate that *R. abnormalis* is able to move the tailspine horizontally and ventrally for defence, steering, or detecting purposes. Interestingly, when we digitally remove parts of the pygidial shield and appendages, two pairs of specialized accessories are exposed (Figure 4D, Figure 5C and Figure 6A,B,G, Appendix A). The slender accessories are inserted near the anterior end of the tailspine. Based on their location and morphology, we speculate that they might have had a reproductive function, but additional data is needed.

*Implications for anamorphic ontogeny*. Compared to other artiopods from Chengjiang (e.g., *Sinoburius*, *Squamacula*, and *Pygmaclypeatus*), *R. abnormalis* is a relatively large animal. Its body size can reach up to nearly 100 mm in length (excluding the antennae and tailspine). The studied specimens cover a wide size range of 21 mm to about 100 mm, with the smaller specimens bearing fewer appendages than the larger specimens (Figure 1E, Figure 2A, Figure 3A, Figure 4D, Figure 5A,C, Figure A1D and Figure A2A,B, Appendix A). Furthermore, the last two pygidial appendages on the studied specimens are clearly smaller than the anterior ones, indicating the later appearance of the more posterior segments (Figure 3A, Figure 5A,C–E and Figure A2A,B, Appendix A). This hints at different developmental stages captured in the studied fossils. Thus, we suggest that *R. abnormalis* undergoes a pattern of anamorphic development during part of its ontogeny, as seen in other non-biomineralized euarthropods from Chengjiang, such as the megacheiran, *Leanchoilia illecebrosa* [16], and the fuxianhuiid, *Fuxianhuia protensa* [29]. Our studies add to the understanding of the developmental mode of Cambrian artiopods, and further support the statement that post-hatching segment addition occurred in the ancestors of Euarthropoda (see [16]).

*Ecological implications*. The new appendicular data on *R. abnormalis* provides some insight into its ecology and functional morphology. Critically, *R. abnormalis* shows the presence of an enlarged protopodite that is transverse elongated. Biomechanical comparisons among the protopodites of Cambrian artiopods and the extant horseshoe crab *Limulus polyphemus* suggest that transversely elongated protopodites are capable of producing substantial forces during mastication, which is commonly associated with a shell-crushing (durophagous) diet [8]. However, the protopodite morphology observed in *R. abnormalis* differs from that of *L. polyphemus*, *Eoredlichia rex* (e.g., [7]) and *Sidneyia inexpectans* (e.g., [30]) in the complete absence of endites. Given that the endites of durophagous species are typically short and robust, their complete absence in *R. abnormalis* may suggest a type of mastication that did not require structures for grinding. However, the limbs of *R. abnormalis* clearly show the presence of robust and paired spinose endites along the ventral side of the endopod podomeres, which were likely employed for grasping, or possibly processing, soft organic matter. We hypothetically reconstruct *R. abnormalis* as a benthic scavenger that mainly fed on soft-bodied organisms and/or organic matter. It is possible that this taxon facultatively consumed small shelly organisms that would require minimal processing prior to digestion, as informed by the proportions of the protopodite that suggest some durophagous capabilities. Unfortunately, there is no information about the digestive tract morphology or gut contents in any of the specimens of *R. abnormalis* studied to date, which complicates reconstructing the autecology of this enigmatic taxon. Finally, the substantial difference of body size (expressed as maximum total length) between *R. abnormalis* (up to 100 mm) compared to its sister taxon *Pygmaclypeatus daziensis* (ca. 20 mm) suggests that despite being close relatives and sharing similar aspects of their overall morphology, these taxa most likely occupied different ecological niches. This strategy would have been advantageous to avoid direct competition for the same food resources, as also suggested in the Chengjiang leanchoiliid *Leanchoilia illecebrosa* [16].

## 5. Conclusions

Micro computed tomography of new material of the non-trilobite artiopod *Retifacies abnormalis* from the early Cambrian Chengjiang allows us to produce a comprehensive redescription of the ventral anatomy of this enigmatic taxon. Previously uncertain aspects of the morphology are resolved, including the absence of eyes, the uniramous nature of the cephalic appendages, and the detailed structure of biramous appendages including proximal exites and exopodites with densely packed paddle-shape lamellae. The smooth protopodite of *R. abnormalis* is a rarity among artiopods and suggests a primary benthic scavenging mode of life. Our phylogentic analysis confirms the sister group relationship of *R. abnormalis* to the diminutive *Pygmaclypeatus daziensis*, and their position as early-branching members of Trilobitomorpha. However, basal trilobitomorph relationships remain sensitive to different analytical methods.

## Figures and Tables

**Table 1 biology-11-01235-t001:** Tomography parameters for the following specimens: YKLP 11430, YKLP 11426, YKLP 11432 and YKLP 11436.

Specimen	Panel	Pixel Size (um)	Detector	Voltage (kV)	Power (W)	Filter	Rotation	Projection	Average Step Rotation
YKLP 11436	Figure 1D	18.03	FP	60	5	no	−120°–+120°	1001 × 7	0.2398°
YKLP 11430	Figure 2A,E	35.92	0.4X	70	6	LE4	−103°–+103°	801 × 2	0.2572°
	Figure 2C,D	17.385	0.4X	70	6	LE4	−103°–+103°	1601	0.1287°
	Figure 2B	9.71	FP	60	5	no	−120°–+120°	1401	0.1713°
	Figure A2, Figure 5B	8.59	0.4X	90	8	LE4	−103°–+103°	1401 × 3	0.1470°
	Figure 5A	7.39	0.4X	90	8	LE4	−103°–+103°	1401 × 2	0.1470°
YKLP 11426	Figure 3D	27.45	FP	60	5	LE4	−120°–+120°	801 × 5	0.2996°
	Figure 3A,B,C	17.01	0.4X	60	5	no	−103°–+103°	1001 × 5	0.2058°
YKLP 11432	Figure 4D	19.83	FP	60	5	no	−120°–+120°	1001 × 3	0.2398°
	Figure 5C,D,E	13.31	0.4X	60	5	no	−103°–+103°	1201 × 3	0.1715°

## Data Availability

Not applicable.

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
