# Peer review of "Ventral Morphology of the Non-Trilobite Artiopod Retifacies abnormalis Hou, Chen & Lu, 1989, from the Early Cambrian Chengjiang Biota, China"

_biology, 2022, doi:10.3390/biology11081235_

Round 1

Reviewer 1 Report

This manuscript “Ventral morphology of the non-trilobite artiopod Retifacies abnormalis Hou, Chen & Lu 1989 from the early Cambrian 3 Chengjiang biota, China” is devoted to elucidating the detailed structural features of the Cambrian non-biomineralized artiopod Retifacies abnormalis. A preservation of these fossils gives an opportunity to use X-ray micro-CT. The authors have achieved excellent results using different techniques of the study. They not only studied the detailed morphology of the animal, but also clarified its systematic position, made conclusions about ontogeny and ecological features. Certainly this article deserves to publish in the journal “Biology”. But there are some remarks.

1. Micro-CT parameters are absent. Only “scanning resolution” is in article. It necessary to show many parameters: current, voltage, step rotation, filters.

2. It is better to write not “scan” or “scanning”, but “tomography”. A scanning is a process of a surface study. Micro-CT is a process of a volumetric study.

3. A reference 22 is cited latter than references 23, 24, 25.

Reviewer 2 Report

No doubt that this is an important contribution to knowledge of early arthropods providing impressive high quality data on the soft anatomy of one of the largest Cambrian animal. However, I find interpretation of the fourth post-antennal appendage not completely convincing. Of course, the authors are free to interpret it as such but, because of its high phylogenetic weight, a more extensive comment supporting their view would be welcome. It quite well may belong to the thorax instead of the cephalic tagma. Somewhat misleading is to refer to the segmented medial appendage as ‘tailspine’, which implies homology (but not analogy) to the rigid chelicerate telson. This structure is rather analogous (if not homologous) to the median caudal filament (or filum terminale) of hexapods. The ‘specialized accessories’ may then correspond to cerci.

Reviewer 3 Report

This is a nice restudy of one of the characteristic arthropods of the Chengjiang fauna, which has been used extensively in phylogenetic analyses. Using modern techniques of photography and scanning, this study produced results which suggest this animal might have been a scavenger or fed on soft-bodied animals. Also, the differentiated  limbs and number of different sizes of animal point to an anamorphic ontogeny for Cambrian artiopods.

Note: line 500, sugges should be suggest
